# Prebiotics and Probiotics: Therapeutic Tools for Nonalcoholic Fatty Liver Disease

**DOI:** 10.3390/ijms241914918

**Published:** 2023-10-05

**Authors:** Alejandra Mijangos-Trejo, Natalia Nuño-Lambarri, Varenka Barbero-Becerra, Misael Uribe-Esquivel, Paulina Vidal-Cevallos, Norberto Chávez-Tapia

**Affiliations:** Médica Sur, Mexico City 14050, Mexico

**Keywords:** probiotics, dysbiosis, NAFLD, gut–liver axis

## Abstract

Alterations in the gut–liver axis and changes in the gut microbiome are among the risk factors for the pathogenesis of non-alcoholic fatty liver disease (NAFLD). These patients show increased bacterial overgrowth in the small intestine and impaired intestinal permeability. Therefore, therapeutic options such as probiotics or prebiotics have been investigated to modulate intestinal microbiota composition to improve NAFLD. Most in vivo and in vitro probiotic studies have focused on reducing hepatic fat accumulation. The beneficial effects of probiotics on NAFLD have been demonstrated in animal models, and the most widely used microorganisms are those of the *Lactobacillus* and *Bifidobacterium* genera. In animal models, probiotics help restore the intestinal microbiota and improve the integrity of the intestinal barrier. This narrative review summarizes published evidence and the likely benefits of probiotics and prebiotics as a therapeutic option for patients with NAFLD.

## 1. Introduction

Non-alcoholic fatty liver disease (NAFLD) is a condition that includes a broad spectrum of histological abnormalities, including isolated steatosis, non-alcoholic steatohepatitis (NASH), and liver fibrosis, which can eventually lead to cirrhosis, liver cancer, and death. NAFLD affects 20–25% of the adult population, and it is estimated that 20% develop steatohepatitis [1].

The pathogenesis of non-alcoholic fatty liver disease (NAFLD) is multifactorial. The multiple-hit hypothesis implicates several factors as causes of this disease. The most notable factors are genetics, obesity, sedentary lifestyle, high-fat diet, insulin resistance, and gut microbiota [2]. According to this theory, there is a vicious circle of fat accumulation in hepatocytes, lipotoxicity, metabolic disorders, inflammation, insulin resistance, and aggravation of metabolic disorders [3].

In addition to the well-known risk factors for the disease, interactions between the gut microbiome, its derived metabolites, the immune system, and the liver contribute to the pathogenesis of NAFLD [4].

This narrative review aims to summarize the evidence for the therapeutic potential of prebiotics and probiotics in treating NAFLD.

## 2. Gut–Liver Axis

Gut microbiota refers to a complex community of microorganisms found in the digestive tract of humans and animals [5]. The gut microbiota is considered a virtual metabolic organ that forms an axis with various extraintestinal organs (kidneys, brain, cardiovascular system, etc.); however, in recent years, the gut–liver axis has attracted the attention of researchers [6]. The gut–liver axis is the anatomical and functional bidirectional relationship between the gut and the liver [4,6].

The portal vein is an anatomical reference point for gut–liver communication and establishes the interaction between the intestinal microbiome and the liver, as it is responsible for transporting gut-derived products to the hepatic circulation [4,7] (Figure 1A). This explains why alterations in the intestinal barrier can lead to the entry of pathogens or their products into the liver, where they can cause or worsen liver diseases [4].

The intestinal barrier (Figure 1B) has an outer layer of mucus, microbiota, and defense proteins such as secretory immunoglobulin A and antimicrobial proteins. The middle layer of the intestinal barrier corresponds to the intestinal epithelial cells, and the inner layer comprises immune cells [8].

The mucus layer is mainly water and contains glycoproteins known as mucins (MUC), produced by goblet cells [8,9,10]. The outer layer of mucus is colonized by microbiota, whereas the inner layer is sterile [4]. The stomach and colon contain both mucus layers; however, only one layer is present in the small intestine. Therefore, the mucus layer of the small intestine is more penetrable by bacteria and toxins. To compensate for the absence of one of these layers in the small intestine, enterocytes, Paneth cells, or immune cells secrete antimicrobial proteins [8,10].

Below the mucus layer, epithelial cells are sealed together by tight junctions (TJs) that release antimicrobial peptides for host defense [4,11]. In addition, the brush border of enterocytes is negatively charged and opposes the negative charge of the microbiota [4].

TJs have two pathways that allow the passage of substances; the first is the “pore” pathway, which is highly selective, and the second is the “leak” pathway, which shows limited selectivity [8].

It has been proposed that in addition to the intestinal epithelial barrier, there is a second intestinal barrier called the gut–vascular barrier (GVB). The GVB comprises of endothelial cells, enteric glial cells, and pericytes that prevent the entry of intestinal microorganisms into the body [11,12]. However, certain bacteria have developed mechanisms to overcome this barrier [4].

In addition to portal circulation, the gut and liver communicate through the flow of hepatic bile [9]. Bile acids are synthesized in the liver and released into the gut, where microbiota further metabolizes them. Microbiota and bile acids have a bidirectional interaction; bile acids affect the composition of the microbiota, which in turn affects bile acid metabolism [4,7,13]. Furthermore, altered bile acid metabolism may promote intestinal dysmotility and systemic inflammation [6].

Liver sinusoidal endothelial cells (LSECs) may act as a hepatic barrier along the liver–gut axis. This is explained by the fact that LSECs are the first in contact with portal-delivered, gut-derived pathogens. LSECs contribute to the uptake and clearance of viruses, bacteriophages, microbial products, and metabolic wastes [14].

Finally, a less-known communication route is a retrograde transit along enteric neurons, which can spread microbes that leak through the intestinal barrier [9].

In the liver, the intestinal barrier must remain intact because increased intestinal permeability facilitates the entry of pathogen-associated molecular patterns (PAMPs) into the portal circulation, thus triggering a proinflammatory cascade and causing hepatic inflammation [4]. PAMPs can also activate stellate cells, which are involved in promoting and progressing liver fibrosis [6] (Figure 2).

One hypothesis suggests that the alteration of the intestinal barrier and the increase in intestinal permeability can produce an inflammatory response, and it is also believed that the intestinal microbiota can modulate this inflammation. This has led to the concept wherein “leaky gut syndrome” and “dysbiosis” are related to each other and could be involved in the pathogenesis of some gastrointestinal and systemic disorders [8].

## 3. Dysbiosis in NAFLD

Trillions of microbes, including bacteria, archaea, viruses, and eukaryotic microbes, colonize the gastrointestinal tract of the human body. The amounts of these microorganisms vary according to the site in the gastrointestinal tract, and the stomach and duodenum have 10–10^3^ bacteria per gram of content; the small intestine (10^4^–10^7^) and large intestine (10^11^–10^12^) are the sites where the highest levels of microbes are found [15]. The dominant phyla in the large intestine are *Firmicutes* and *Bacteroidetes*; however, other phyla, such as *Actinobacteria* and *Proteobacteria*, are also present [5].

The colon also harbors essential pathogens, such as *Escherichia coli* (*E. coli*), *Campylobacter jejuni*, *Salmonella enterica*, *Vibrio cholerae*, and *Bacteroides fragilis*; however, these pathogens usually are found at deficient levels (<0.1% of the gut microbiome) [15]. It is estimated that there are approximately 10^5^–10^6^ fungal cells per gram of feces. The main fungal phyla in the intestines are *Ascomycota* and *Basidiomycota* [13].

These microbes interact with each other and cells in the human body in complex and poorly understood ways. These interactions affect the host’s metabolism, shape immunity, and facilitate digestion and nutrient absorption, among other functions [16].

These microorganisms interact with one another through mutualism, commensalism, and competition [13]. Furthermore, bacteria in a healthy gut are in homeostatic balance with their host and contribute to maintaining a healthy state. Dysbiosis is an imbalance or change in bacterial content or metabolic function [15].

Several factors can modify the composition and function of the intestinal microbiota, including the host’s genetics, diet, age, antibiotics, and smoking. In addition, changes in the intestinal microbiota may contribute to developing diseases, such as NAFLD [5].

A decrease in microbial gene richness (MGR) is associated with a proinflammatory state, adiposity with abdominal distribution, and a propensity for metabolic alterations, all related to the pathophysiology of NAFLD. People with a low MGR show an increase in bacteria capable of synthesizing lipopolysaccharides, which is related to insulin resistance and an adverse lipidomic profile, a part of the pathophysiology of NAFLD [17]. Some studies have identified metabolites derived from gut bacteria that may be involved in developing hepatic steatosis [13].

Patients with NAFLD and NASH show increased numbers of *Bacteroidetes* and changes in the presence of *Firmicutes*, resulting in a decreased F/B ratio. However, owing to the different molecular methods used to identify bacteria, F/B ratios can vary and produce inconsistent results. Patients with NAFLD have a higher proportion of *Clostridium*, *Anaerobacter*, *Streptococcus*, *Escherichia*, and *Lactobacillus* and a lower proportion of *Oscillibacter*, *Flavonifaractor*, *Odoribacter*, and *Alistipes* spp. (Table 1) [18]. They show an increase in *Proteobacteria* and *Enterobacteriaceae*, along with a decrease in *Rikenellaceae* and *Rumminoccaceae* [17].

There is evidence of a decrease in viral diversity and the proportion of bacteriophages in patients with NAFLD. The gut microbiome is affected in the early stages of the disease and is altered at later stages [13].

In advanced fibrosis (F3–F4), an increase in Gram-negative microorganisms, a decrease in *Firmicutes*, and a greater number of *Proteobacteria* have been documented. *Escherichia coli* and *Bacteroides vulgatus* are the most abundant bacteria at the species level. Patients with NAFLD cirrhosis show more abundance of species within the *Enterobacteriaceae* family and *Streptococcus* genera [17].

Individuals with NAFLD exhibit increased bacterial overgrowth in the small intestine and impaired intestinal permeability [9]. Therefore, numerous therapeutic options have been proposed to modulate the composition and function of intestinal microbiota, including probiotics, prebiotics, and fecal microbiota transplantation (FMT) [5].

## 4. Prebiotics and Probiotics in NAFLD

Probiotics are live microorganisms that can confer a health benefit on the host; the main genera of probiotics studied are *Lactobacillus* and *Bifidobacterium*. In turn, prebiotics are non-viable food components associated with microbiota modulation and can confer a health benefit on the host. Prebiotics consist primarily of polysaccharides (inulin, cellulose, hemicellulose, pectins, and resistant starch) and oligosaccharides (fructooligosaccharides, galactooligosaccharides, isomaltooligosaccharides, xylooligosaccharides, lactulose, and soy oligosaccharides), which stimulate the growth of beneficial bacteria. The most studied prebiotics in patients with NAFLD are fructooligosaccharides. Finally, synbiotics are a mix of probiotics and prebiotics [19,20,21].

Different mechanisms of action have been proposed by which probiotics and prebiotics could benefit patients with NAFLD. These exert beneficial effects by altering the composition of the microbial flora. Probiotics can act in different target organs by producing antimicrobial peptides, reducing intestinal permeability, or preventing the translocation of bacterial products [12]. *Lactobacillus* and *Bifidobacterium* have been reported to be associated with ß-glucuronidase inhibition [19]. At the same time, *Bifidobacterium* protects against proinflammatory cytokine secretion and intestinal barrier dysfunction [22].

Probiotics have positive effects on inflammatory liver damage mediated by c-Jun N-terminal kinase (JNK) and Nuclear Factor kappa B (NF-κB), which was correlated with Tumor Necrosis Factor-alpha (TNF-α) regulation and insulin resistance [6].

Studies using prebiotics in NAFLD are limited compared to probiotics [21]. Prebiotics can selectively promote the proliferation and activity of intestinal microbes. Animal models have shown that prebiotic supplementation reduces the fatty acid synthesis pathway, which may decrease fructose-induced hepatic triglyceride (TG) accumulation; this could be due to reduced gene expression of enzymes that regulate hepatic lipogenesis, such as acetyl Co-A carboxylase and fatty acid synthase. In addition, oligofructose modifies intestinal microbiota in favor of *Bifidobacterium*, which improves mucosal barrier function and reduces the level of endotoxins [12].

Administration of prebiotics could reduce liver inflammation through a glucagon-like peptide 2-dependent effect on the intestinal barrier; however, further studies are still needed to determine the mechanisms by which they could be beneficial in NAFLD [6].

Finally, postbiotics are a broad range of bioactive molecules, including non-viable microbial cells, cell compounds, and any soluble products or metabolic by-products resulting from microorganisms, which confer a health benefit on the host. Postbiotics have been recognized to mimic the functions and activities of probiotics. These have not been directly evaluated in patients with NAFLD; however, their antioxidant, anti-obesogenic, anti-inflammatory, and anti-adipogenesis effects have been studied, as well as their action on insulin resistance [23].

## 5. Data from In Vitro Models

Hepatic fat metabolism may be influenced by the presence of commensal bacteria and potentially by probiotics; however, the mechanisms by which probiotics act on the liver are unclear. Several in vitro studies have been conducted to support in vivo findings regarding probiotic capacities in NAFLD models. The probiotic activities of various isolated bacterial strains were systematically studied and demonstrated in vitro, along with their growth characteristics, stress resistance, intestinal colonization ability, and antioxidant and antagonistic activities against pathogens [24,25,26]. Specifically, MIYAIRI 588-a butyrate-producing probiotic showed that treatment with sodium butyrate (NaB) activated AMPK (5′ adenosine monophosphate-activated protein kinase) and AKT (protein kinase B), thus enhancing the expression of nuclear factor erythroid 2-related factor [27,28].

Most studies on probiotic strains for NAFLD have focused on their ability to reduce lipid accumulation significantly. In an in vitro model of HepG2 cells treated with oleic acid and cholesterol, *Lactiplantibacillus plantarum* (*Lactobacillus plantarum*) AR113 and *Lacticaseibacillus casei* (*Lactobacillus casei*) pWQH01, probiotic strains with high bile salt hydrolase activity, showed significantly reduced lipid accumulation, lipid content, and total cholesterol levels. It targets AMPK-mediated fatty acid synthesis, thus inhibiting SREBP-1c (sterol regulatory element-binding protein 1c), ACC (Acetil Coenzima), and fatty acid synthase (FAS) expression. *L. plantarum* and *L. casei* can improve steatosis in vitro in a bile salt hydrolase-dependent manner [29].

To test whether the fatty acid consumption capacity of *Lacticaseibacillus rhamnosus* (*Lactobacillus rhamnosus*) GG affects cellular fat accumulation in vitro, a cloned cell line from intestinal CaCo-2 cells was employed. *Lacticaseibacillus rhamnosus GG* consumed exogenous oleic acid more than the other *Lactobacillus* strains. Specifically, *Lacticaseibacillus rhamnosus GG* preferred fatty acids as its substrate during cultivation, and this probiotic strain reduced oleic acid-induced lipid accumulation in intestinal cells by limiting the exogenous oleic acid source. This suggests a novel mechanism by which *Lacticaseibacillus rhamnosus GG* consumes intestinal fatty acids and protects against the initial stages of NAFLD development [30]. *Latilactobacillus sakei* (*Lactobacillus sakei*) MJM60958 was tested in vitro on HepG2 cells stimulated with oleic acid and cholesterol. This strain significantly inhibited lipid accumulation [31].

Interestingly, the use of *lactobacilli* probiotic strains to reduce blood cholesterol levels has been extensively reported regarding gene regulation responsible for the intestinal transport of cholesterol and homeostasis of cholesterol in the liver [32,33]. Screening for in vitro cholesterol-lowering properties is essential for selecting bacterial strains for further in vivo probiotic investigations [34]. Thirty-four strains of *Bifidobacteria* were assayed in vitro for their ability to assimilate cholesterol and bile salt hydrolases (BSH) against glycolic and taurodeoxycholic acids (GCA and TDCA). Administration of this probiotic significantly reduced total cholesterol and low-density cholesterol levels [35].

The cholesterol-lowering properties of lactobacilli could be mediated through the AMPK pathway, specifically through the phosphorylation of AMPK, leading to reduced expression of HMG-CoA reductase (3-hidroxi-3-metil-glutaril-CoA reductase) [34].

## 6. Data from In Vivo Models

The beneficial effects of probiotics on the liver have been mainly demonstrated in animal models of NAFLD. Probiotic preparations contain one or more microbial strains, the most-used microorganisms within the genera *Lactobacillus* and *Bifidobacterium*. Owing to the variety of mechanisms exerted by each strain or a combination of strains, the choice of strain is crucial for therapeutic success.

### 6.1. Lactobacillus

*Lacticaseibacillus rhamnosus* GG increases the expression of hepatic FGF21 and adiponectin, resulting in the upregulation of SphK2 (Sphingosine kinase 2) and inactivation of PP2AC (protein phosphatase 2 catalytic subunit alpha), leading to reduced carbohydrate-responsive, element-binding protein (ChREBP) activity and fructose-induced reversal of NAFLD [36].

The supply of *Limosilactobacillus reuteri* (*Lactobacillus reuteri*) induces IL-22 secretion, which reduces hepatic triglyceride levels in a diet-induced obesity model [37]. After treatment with the probiotic Eosinophil-*Lactobacillus*, the mRNA and protein expressions of FXR (receptor farnesoid X) and FGF15 (fibroblast growth factor 15) increased, indicating that Eosinophil-*Lactobacillus* may affect bile acid metabolism by upregulating the expression of the FXR/FGF15 pathway [38].

The administration of *Latilactobacillus sakei* MJM60958 significantly reduced body and liver weights and controlled the levels of aspartate transaminase (AST), aspartate aminotransferase (ALT), TG, uric acid (UA), and urea nitrogen (BUN) in the blood, which are characteristic of NAFLD. MJM60958 treatment also decreased liver tissue steatosis scores, serum leptin, and interleukin levels and increased adiponectin levels. In addition, it significantly reduced the expression of some genes and proteins related to lipid accumulation, such as sterol regulatory element-binding protein 1 (SREBP-1), acetyl-CoA carboxylase (ACC), and FAS. It also increases the expression of proteins related to lipid oxidation, such as carnitine palmitoyl transferase 1a (CPT1A) and peroxisome proliferator-activated receptor alpha (PPARα) [31].

Oral treatment with *Ligilactobacillus salivarius* (*Lactobacillus salivarius*) SNK-6 in laying hens reduced liver fat by regulating lipid metabolism through the miR-130a-5p/MBOAT2 (microRNA-130a-5p/membrane-bound O-acyltransferase domain containing 2) pathway, including FAS, SREBP-1, FABP4 (fatty acid-binding protein 4), and PPARγ (peroxisome proliferator-activated receptor gamma) genes. It also lowered serum total cholesterol and triglyceride levels, as well as the activities of AST and ALT [25].

Treatment with *Lactiplantibacillus plantarum* ZJUIDS14 promoted the uptake and biosynthesis of fatty acids and triglycerides through increased expression of SREBP-1C (Sterol Regulatory Element-Binding Protein-1c), fatp2 (fatty acid transport protein 2), and fatp5 (fatty acid transport protein 5). It also increases fatty acid β-oxidation by activating PPARα (peroxisome proliferator-activated receptor alpha) in high-fat diet (HFD)-fed mice. In contrast, *Lactiplantibacillus plantarum* ZJUIDS14 increases DRP1 (dynamin-related protein 1) levels, suggesting that the probiotic triggers mitochondrial fission [39].

*Limosilactobacillus reuteri* prevents NAFLD progression through gut dysbiosis and the phospho-Akt/mammalian target of rapamycin/LC-3 II (p-AKT/mTOR/LC-3II) pathways, thus improving insulin resistance, increasing oxidation and subsequently decreasing liver weight and blood pressure fat accumulation [40].

Treatment with *Limosilactobacillus fermentum* (*Lactobacillus fermentum*) CQPC06 decreases intestinal permeability, thereby reducing lipopolysaccharide (LPS) content and inhibiting abnormal inflammatory cytokine production, alleviating abnormal lipid metabolism and fat accumulation by upregulating protein expression and CPT1 (carnitine palmitoyl transferase type 1), PPARα, LPL (Lipoprotein lipase), and CYP7A1 (Cholesterol 7 alpha-hydroxylase) genes and downregulating PPARγ and C/EBP-α (CCAAT enhancer binding protein α) expression. *Limosilactobacillus fermentum* CQPC06 is an antioxidant that reduces ROS levels of reactive oxygen species in the liver [41].

*Lactiplantibacillus plantarum* NA136 increased the relative proportions of *Alistipes*, *Enterorhabdus*, *Desulfovibrio*, and *Prevotella* in the gut microbiota, thereby improving gut barrier integrity by normalizing tight junction protein expression levels and inhibiting the translocation of gut bacteria. It also decreases the levels of inflammatory cytokines such as interleukin (IL)-1β, IL-6, and tumor necrosis factor α (TNF-α), counteracts hepatic lipid metabolic disorders, alleviates insulin resistance, and activates antioxidant responses in NAFLD [42].

*Lactococcus lactis* (*Lactobacillus lactis*) sp. cremoris ATCC 19257 decreases hepatic resolvin E1 and hydroxy-octadecadienoic acid levels and reduces liver inflammation [43].

### 6.2. Probiotic Mixture

VSL#3 *Lactobacillus* has anti-inflammatory activity, as it reduces Jun N-terminal kinase and nuclear factor (NF-κB) activity in the liver, thus reducing cyclooxygenase 2 and inducible nitric oxide synthase (iNOS) expression. In addition, it decreases fatty acid oxidation and TNF-α activity, along with the levels of alanine aminotransferase in the serum [44].

*Bifidobacteria* L66-5, L75-4, M13-4, and FS31-12 reduced serum and hepatic triglyceride levels, whereas only *Bifidobacteria* L66-5 and *Bifidobacteria* FS31-12 significantly decreased their levels in the liver. A mixture of *Bifidobacterium breve* and *Lacticaseibacillus paracasei* (*Lactobacillus paracasei*) decreases hepatic triglyceride content and reduces steatosis by decreasing serum LPS [45].

Intervention with a probiotic mixture containing *Bifidobacterium infantis*, *Lactobacillus acidophilus*, and *Bacillus cereus* counteracts HFD-induced dysbiosis by upregulating intestinal TJ protein expression, thus improving gut integrity. In addition, these probiotics delay NAFLD by inhibiting the LPS-TLR4 (lipopolysaccharide/Toll-like receptor 4) signaling pathway. It also improves liver inflammation by decreasing IL-18 (interleukin-18) levels in the serum, thereby improving the degree of hepatic steatosis [46].

After administration of *Lactobacillus acidophilus*, *Lacticaseibacillus casei*, *Limosilactobacillus reuteri*, and *Bacillus coagulans*, the serum levels of TG, glucose, and ALT, as well as inflammatory and oxidative stress markers, improve significantly. In the liver, TG levels improve due to the expression and activation of the PPARα receptor [47].

When a combination of *Lactobacillus bulgaricus*, *Lacticaseibacillus casei*, *Lactobacillus helveticus*, and *Pediococcus pentosaceus* KID7 is administered, cholesterol, TNF-α, IL-1β, and IL-6 levels significantly improve, in addition to improvements in intestinal leaks. It also decreases the number of macrophages in the liver [48].

Administering probiotics composed of *Bifidobacterium bifidum* V and *Lactiplantibacillus plantarum* X improves cholesterol and TG levels in the liver as well as serum levels of alanine transaminase, aspartate transaminase, free fatty acids, triglycerides, and cholesterol [49].

Supplementation with Lactoferrin-expressing bacteria, such as *Lactobacillus delbrueckii* (BCRC 14008), *Lactiplantibacillus paraplantarum*, *Lactobacillus gasseri*, *Lacticaseibacillus rhamnosus*, *Bifidobacterium angulatum* (ATCC 27535), and *Bifidobacterium breve* (BCRC12584), improves the serum lipid profile by reducing cholesterol and TGs, increasing high-density lipoprotein (HDL), and decreasing hepatic steatosis [50].

Treatment with *Bifidobacterium adolescentis* and *Lacticaseibacillus rhamnosus* reversed the low-density lipoprotein cholesterol/high-density lipoprotein cholesterol (LDL-C/HDL-C) ratio, decreased serum LPS, elevated TJ protein transcription, and increased short-chain fatty acid levels, thus reducing inflammation [51].

In general, probiotics in animal models help restore the intestinal microbiota, which improves the integrity of the intestinal barrier. They also decrease blood lipid levels and alanine transaminase and aspartate transaminase activity. They reduce hepatic fat accumulation and inflammation, thereby mitigating liver pathology (Table 2).

### 6.3. Prebiotics

Among the prebiotics, pectin has been shown in animal models to improve hepatic steatosis and decrease serum ALT values and inflammatory cytokines; however, in humans, the administration of large amounts is complicated due to its poor palatability and its side effects, which include increased abdominal discomfort and intestinal pain [52,53,54,55].

Concerning oligosaccharides, chitosan was found to have a liver-protective effect and a protective effect on tight junctions in the gut by enriching for a higher abundance of *Coprococcus* and reducing *Mucis pirillum*. At the same time, chitosan regulates pathways related to lipids and inhibits the expression of free fatty acid synthesis and pro-inflammatory genes. However, human clinical trials are needed to corroborate this information [56].

Cyclodextrin is another prebiotic without clinical trials in humans which has been reported to reduce lipid accumulation in the liver and decrease liver inflammation in animal models [57].

## 7. Clinical Use of Probiotics for NAFLD

Various therapeutic targets regulate gut dysbiosis, including antibiotics, prebiotics, probiotics, synbiotics, and fecal transplantation [15]. Studies have mainly focused on using different species of *Lactobacillus*, *Bifidobacterium*, and *Streptococcus* as probiotics and fructooligosaccharides (FOS) as prebiotic [19]. Some major randomized clinical trials of probiotics and prebiotics conducted in the last five years in patients with hepatic steatosis are presented in Table 3.

A meta-analysis of 21 studies (1252 participants) reported that the administration of probiotics and synbiotics was associated with a reduction in liver stiffness measurement (by elastography) and steatosis grade (by ultrasound) [65]. Another meta-analysis of nine studies demonstrated that probiotic therapy reduced serum levels of ALT, AST, and total cholesterol compared to those in the control group [66]. Khan et al. observed decreased serum transaminase levels following probiotic treatment [67]. A reduction in serum GGT levels has also been observed using some probiotics and synbiotics; however, other studies do not support this [19].

Li et al. performed a meta-analysis that showed an improvement in the biomarkers of energy metabolism (glucose, insulin, and total cholesterol) using probiotics, prebiotics, and synbiotics in a population with NAFLD [68]. Huang et al. reported similar results in their review, observing an improvement in insulin resistance and a decrease in serum levels of TG, AST, and GGT [69]. A recent meta-analysis by Zhou et al. included 21 studies (1037 participants) and demonstrated that administering probiotics for a minimum period of 12 weeks improved serum glucose, TG, and GGT levels [70].

However, a comprehensive meta-analysis concluded that more studies are needed to demonstrate the effects of probiotics, prebiotics, and synbiotics in patients with NAFLD [71]. This is due to the high heterogeneity of the studies and small sample sizes in the trials; therefore, no clear recommendations can be made [16].

Probiotics, prebiotics, and synbiotics are first-generation microbial therapies. Engineered bacteria have been classified as next-generation microbiome therapies designed to produce beneficial metabolites or toxic products. The latter requires further studies in patients with NAFLD to propose its use, as it has only been studied in animal models [16].

Direct FMT is a safe and widely used treatment for gastrointestinal infections caused by *Caenorhabditis difficile*. FMT restores the commensal gut microbiota and repairs “indirect mechanisms” of resistance to colonization to avoid the penetration/translocation of potential pathogens. Therefore, FMT has been suggested for other alterations in the gut microbiome, such as those found in NAFLD [72].

There are studies on FMT in patients with cirrhosis and alcoholic hepatitis; however, studies evaluating the safety of fecal transplantation for NAFLD treatment are lacking [16].

The diet also modulates the gut microbiota. For example, a diet rich in fiber and monounsaturated and polyunsaturated fatty acids promotes healthy microbiota. Therefore, the Mediterranean, vegetarian, and vegan diets have been shown to improve dysbiosis by increasing *Bifidobacterium*, *Prevotella*, and *Faecalibacterium prausnitzii* while reducing *E. coli* and other gram-negative bacteria, which is beneficial in patients with NAFLD; however, further studies are needed [19].

## 8. Discussion and Future Directions

The prevalence of NAFLD is increasing worldwide; therefore, it is crucial to identify novel therapeutic targets. Given the existing research conducted in animal models, we now have the tools to understand the likely effects of probiotics on NAFLD. Animal models have shown that the use of probiotics, specifically *Lactobacillus* and *Bifidobacterium* genera, can improve intestinal dysbiosis, improve liver steatosis by histology, reduce fatty acid content, decrease serum transaminase levels, and decrease inflammatory cytokine levels. Additionally, some studies have reported that probiotics improve serum lipid levels, regulate the production of short-chain fatty acids, and improve metabolic parameters, which could contribute to improving hepatic steatosis. However, most of these studies have been conducted in vivo; therefore, more randomized clinical trials in humans are needed to recommend its use. Few published clinical trials have used mixtures of different strains of probiotics, making it difficult to reproduce such findings. In addition, most studies involve a short duration of probiotic administration and a short follow-up period; therefore, it is also necessary to carry out studies with a longer duration. Another research direction is prebiotics, synbiotics, and fecal transplantation, which have been investigated in very few studies but have shown favorable results in patients with NAFLD.

## 9. Conclusions

The information pooled in this narrative review shows the prebiotics and probiotics therapeutic potential in NAFLD treatment.

## Figures and Tables

**Figure 1 ijms-24-14918-f001:**
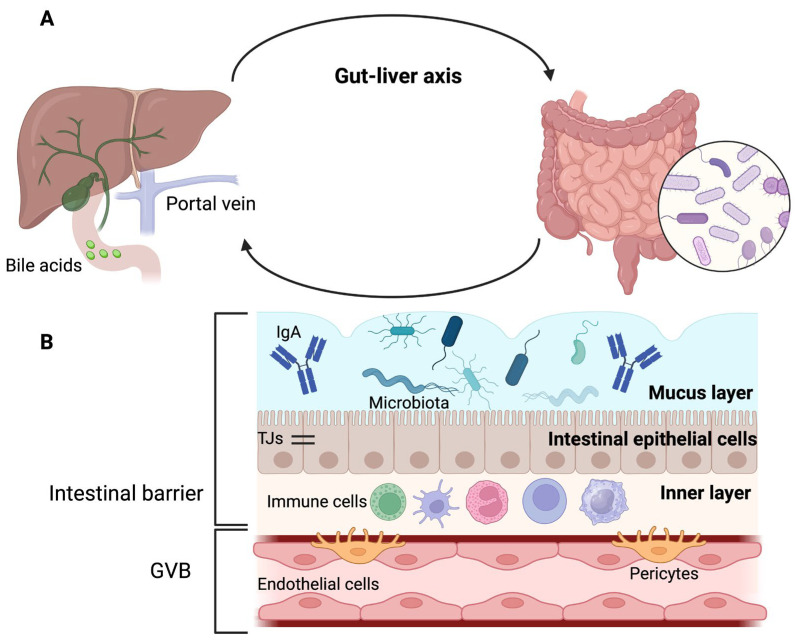
Gut–liver axis and intestinal barrier. (**A**) Gut, portal circulation, liver, and bile duct are connected anatomically. (**B**) The intestinal barrier has three layers: the outer layer is made up of mucus, microbiota, and defense proteins such as immunoglobulin A (IgA); the middle layer corresponds to intestinal epithelial cells that are sealed together by tight junctions (TJs), and the inner layer is composed of immune cells. The gut–vascular barrier (GVB) constitutes a second protective barrier. Adapted from ref. [4]. Created by BioRender (accessed on 29 August 2023).

**Figure 2 ijms-24-14918-f002:**
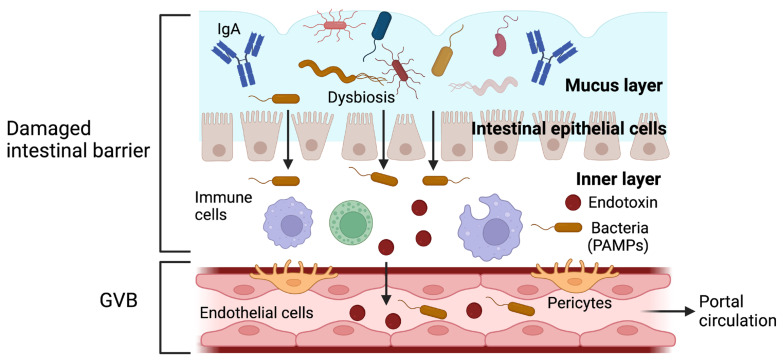
The function of the intestinal barrier after being damaged. Alteration of the intestinal barrier increases in intestinal permeability and facilitates the translocation of bacteria and endotoxins, which can damage the gut–vascular barrier (GVB). The entry of pathogens and pathogen-associated molecular patterns (PAMPs) into the portal circulation triggers an inflammatory response. Gut-derived PAMPs can bind to specific toll-like receptors in the liver and activate the proinflammatory pathways, which result in hepatic inflammation. Adapted from ref. [4]. Created by BioRender (accessed on 29 August 2023).

**Table 1 ijms-24-14918-t001:** Microbiota in healthy subjects vs. NAFLD according to genus.

Increased	Diminished
*Lactobacillus*	*Alistipes* spp.
*Robinsoniella*	*Prevotella*
*Roseburia*	*Odoribacter*
*Dorea*	*Flavonifractor*
*Anaerobacter*	*Oscillibacter*
*E. coli*	
*Clostridium XI*	
*Streptococcus*	

**Table 2 ijms-24-14918-t002:** Probiotics used in animal models.

Probiotic	Experimental Model	Therapy Duration	Results	Reference
Lactobacillus				
*Lacticaseibacillus rhamnosus GG* (LGG)	FGF21 * knockout and C57BL/6 wild typ mice, fed 30% fructose.	4 weeks	LGG administration reverses the reduced FGF21 expression, increases adipose production of ADPN *, and reduces hepatic fat accumulation and inflammation in the WT * mice but not in the KO * mice.	[36]
*Limosilactobacillus reuteri*	Diet-induced obese mouse model	8 weeks	Induces IL-22 * secretion, which reduces hepatic triglycerides.	[37]
Eosinophil-*Lactobacillus*	High-fat-diet rat model	8 weeks	Lowers blood lipid levels, improves liver pathology, improves gut microbiota dysbiosis, and increases bile acid receptor expression through the gut microbiota/FXR/FGF15 * signaling pathway.	[38]
*Latilactobacillus sakei* MJM60958	High-fat-diet mouse model	12 weeks	It improves the metabolism of fatty liver and reduces NAFLD, since it decreases the expression of genes and proteins related to the synthesis of hepatic lipids and increases the levels of genes and proteins related to fat oxidation.	[31]
*Ligilactobacillus salivarius* SNK-6	Xinyang black-feather laying hens NAFLD model	12 weeks	Inhibits fat deposition in the liver and decreases serum triglyceride levels, as well as alanine transaminase and aspartate transaminase activities.	[25]
*Lactiplantibacillus plantarum* ZJUIDS14	High-fat-diet male C57BL/6 mice	12 weeks	Mitigates hepatic steatosis by modulating the balance of the intestinal microbiota and the integrity of the intestinal barrier; strengthens mitochondrial function, and increases fatty acid oxidation.	[39]
*Limosilactobacillus reuteri* DSM 17938	High-fat-diet Male SD * rats	4 weeks	Decreases hepatic steatosis, reduces alanine transaminase, aspartate transaminase, glucose, insulin, cholesterol, triglycerides, and LDL * levels along with an increase in HDL * levels. In addition, an increase in lipid peroxidation and a decrease in hepatic reserves of GSH *.	[40]
*Limosilactobacillus fermentum* CQPC06	High-fat and fructose diet male C57/BL6J mice	8 weeks	Downregulates cholesterol, triglycerides and LDL * levels in serum and liver, and it upregulates the concentration of HDL *.	[41]
*Lactiplantibacillus plantarum* NA136	High-fat and fructose diet male C57/BL6J mice	16 weeks	It inhibits the growth of harmful bacteria, improves the integrity of the intestinal barrier, and reduces inflammatory responses.	[42]
*Lactococcus lactis* Subspecies *cremoris*	High-fat, high-carbohydrate diet female C57BL/6 mice	16 weeks	Improves metabolic parameters and liver adiposity.	[43]
Probiotic mixture				
VSL#3 *Lactobacillus*	High-fat-diet ob/ob mice	4 weeks	VSL#3 improves liver histology, reduces liver fatty acid content, and decreases serum alanine aminotransferase levels.	[44]
*Bifidobacteria* L66-5, L75-4, M13-4 and FS31-12	Monosodium-glutamate-diet Wistar male rats	12 weeks	Reduces hepatic steatosis.	[45]
*Bifidobacterium infantis*, *Lactobacillus acidophilus*, and *Bacilluscereus*	High-fat-and-fructose diet male SD * rats	12 weeks	Restores the intestinal flora microecosystem and upregulates the expression of occludin, which inhibits the entry of bacteria or endotoxins into the blood circulation and decreases the expression of TLR4 * in the liver; therefore, it reduces the hepatic and systemic inflammatory responses.	[46]
*Lactobacillus acidophilus*, *Bacillus coagulans*, *Lacticaseibacillus casei*, *Limosilactobacillus reuteri*	NAFLD-induced male SD * rats		Preserves lipid profiles and reduces hepatic steatosis.	[47]
*Pediococcus pentosaceus* KID7 and *Lactobacillus bulgaricus*	Western diet-fed male C57BL/6J mice	9 weeks	It improves intestinal dysbiosis by modulating the intestinal microbiome, and it decreases inflammatory cytokines.	[48]
*Bifidobacterium bifidum* V and *Lactiplantibacillus plantarum* X	High-fat-diet-fed male C57BL/6N mice	Not specified	Reduce serum levels of alanine transaminase, aspartate transaminase, free fatty acids, triglycerides, and cholesterol and ameliorates cholesterol and triglyceride levels in the liver.	[49]
*Lactobacillus delbrueckii* (BCRC 14008), *Lactiplantibacillus paraplantarum*, *Lactobacillus gasseri*, *Lacticaseibacillus rhamnosus*, *Bifidobacterium angulatum* (ATCC 27535), and *Bifidobacteriumbreve* (BCRC12584)	High-fat-diet-fed C57BL/6 mice	4 weeks	Improves hepatic steatosis by improving the serum lipid profile.	[50]
*Bifidobacterium adolescentis* and *Lactobacillus rhamnosus*	High-fat-high-cholesterol-diet-fed C57BL/6J mice	23 weeks	Regulates the production and concentration of short-chain fatty acids through interactions with the intestinal microbiota, which regulates hepatic steatosis.	[51]
*Bifidobacterium adolescentis* and *Lacticaseibacillus rhamnosus*	Monosodium-glutamate-diet-induced NAFLD model in rats	12 weeks	Reduces the accumulation of hepatic lipids, the proinflammatory cytokines.	[45]

* FGF21 (fibroblast growth factor 21); ADPN (adiponutrin); WT (wild-type); KO (Knockout); IL-22 (interleukin-22); FXR (receptor farnesoid X); FGF15 (fibroblast growth factor 15); LDL (low-density lipoprotein cholesterol); HDL (high-density lipoprotein cholesterol); GSH (glutathione); SD (Sprague Dawley); TLR4 (Toll-like receptor 4).

**Table 3 ijms-24-14918-t003:** Probiotics and prebiotics used in recent clinical trials.

Probiotic/Prebiotic	Therapy Duration	Effects on NAFLD	Trial/Reference
Probiotic capsule with 5 × 10^9^ CFU five bacterial strains (*Lactobacillus casei*, *Lactobacillus rhamnosus*, *Lactobacillus acidophilus*, *Bifidobacterium longum*, and *Bifidobacterium breve*) versus placebo	12 weeks	↓ Serum ALT, AST, ALP, and GGT levels	[58]
Multi-strain probiotic sachet containing different species of *Lactobacillus* and *Bifidobacterium* at a concentration of 30 billion CFU twice a day (*Lactobacillus acidophilus*, *Lactobacillus casei*, *Lactobacillus lactis*, *Bifidobacterium bifidum*, *Bifidobacterium infantis*, *Bifidobacterium longum*) versus placebo	6 months	There was no improvement in liver fibrosis parameters	[59]
Probiotic concentrate of eight different strains of bacteria (*Streptococcus thermophilus*, *Bifidobacterium breve*, *Bifidobacterium longum*, *Bifidobacterium infantis*, *Lactobacillus acidophilus*, *Lactobacillus plantarum*, *Lactobacillus paracasei*, and *Lactobacillus bulgaricus*) versus placebo	10 weeks	No significant improvements in markers of liver injury	[60]
Two multi-strain probiotic capsule three times a day (each capsule contains 112.5 billion bacteria: *Lactobacillus paracasei*, *Lactobacillus plantarum*, *Lactobacillus acidophilus*, *Lactobacillus delbrueckii subsp. bulgaricus*, *Bifidobacterium longum*, *Bifidobacterium infantis*, *Bifidobacterium breve*, and *Streptococcus thermophilus*) versus placebo	12 months	Improvement in liver histology and reduction in steatohepatitis	[61]
Mixture of six probiotic agents (*Lactobacillus acidophilus*, *Lactobacillus rhamnosus*, *Lactobacillus paracasei*, *Pediococcus pentosaceus*, *Bifidobacterium lactis y B. breve*) versus placebo	12 weeks	Reduction in intrahepatic fat and body weight in obese patients with NAFLD	[62]
Oligofructose (8 g/day for 12 weeks followed by 16 g/day for 24 weeks) versus placebo	36 weeks	Reduction in hepatic steatosis	[63]
Metronidazole 400 mg twice daily for a week then inulin administration at 4 g twice daily	12 weeks	↓ Serum ALT levels	[64]

ALP (alkaline phosphatase); ALT (alanine transaminase); AST (aspartate aminotransferase); GGT (gamma-glutamyl transferase); CFU (colony-forming units). ↓: Decrease.

## Data Availability

No new data were created nor generated in this manuscript. Data sharing is not applicable to this article.

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
