# Peer review of "Prebiotics and Probiotics: Therapeutic Tools for Nonalcoholic Fatty Liver Disease"

_ijms, 2023, doi:10.3390/ijms241914918_

Round 1
Reviewer 1 Report
This paper presents an interesting theoretical study, with beneficial implications in the pathology of NAFLD, through the use of certain probiotic and prebiotic preparations to improve the health of patients. The authors described the main characteristics of the gut microbiota, the bidirectional relationship between the gut and the liver, the dysbiosis in NAFLD, the data obtained from in vitro and in vivo models and the clinical use of probiotics for NAFLD. The research described by the authors demonstrated that favorable results were recorded in patients with NAFLD, following the administration of probiotic and prebiotic preparations.
1. The presentation of the conducted research should be a little more organized; the main types of probiotic and prebiotic preparations studied, the main microbial genera existing in the intestinal microflora should be emphasized; it should be specified if the mechanism of action of probiotics and prebiotics in NAFLD pathology is known, if there are articles describing this mechanism.
2. Prebiotic preparations are relatively little discussed in this paper, being described only in section 6. It would have been interesting to show which are the main categories of prebiotics studied, what role they have for the development of probiotic bacteria in the intestine. It should also be mentioned if research has been carried out regarding the effect of postbiotic preparations on NAFLD.
3. In most of the text, the scientific name of the microorganisms should be checked and correctly written (with italics).
4. Line 15, 189 – the scientific name of the genus is Bifidobacterium, and Bifidobacteria refers to a certain type of bacteria
5. Table 1 - "E. Coli” is correct E. coli; line 127 "Alistipes spp" – correct is Alistipes spp
6. - symbiotic (line 296, 351) or synbiotic (line 306)?
7. In section “7. Conclusions and future directions”, before the future trends, the main effects of the probiotic and prebiotic preparations found in the specialized literature and described previously should be clearly presented.
Reviewer 2 Report
Review to the manuscript of a review paper “Prebiotics and probiotics: Therapeutic tools for nonalcoholic fatty liver disease” by Alejandra Mijangos-Trejo et al. submitted to International Journal of Molecular Sciences.
The manuscript is a comprehensive review of probiotics or prebiotics that have been confirmed to modulate the composition of the intestinal microbiota to relieve non-alcoholic fatty liver disease. The manuscript is not sufficiently illustrated, as no scheme or figure is presented. The methodology for conducting the review is not explained. These aspects make the comparison of different studies more difficult.
The occurrence of the disease has been constantly increasing and therefore therapeutic tools for preventing, supporting the treatment and relieving symptoms has significant importance. Microbiome has enormous role in the health and disease of the organism and gut-liver axis has some impact. Therefore, the field of pre- and probiotics have been very extensively studied and reviewed. From the recent 5 years, several systematic reviews on the exactly same subject are available in the international scientific journals therefore this fact does not warrant the publication. The manuscript does not add a new approach or significant amount of additional information.
The manuscript uses correct academic English and is generally easily readable. There are many formatting issues (hyphen usage, italics, references) in the manuscript.
Reviewer 3 Report
Dear authors,
I appreciate the opportunity to review this document.
In making an overall reading, I see some important points for improvement/modification in order to consider this manuscript for publication.
This is a current and developing topic. The use of prebiotics and probiotics as a therapeutic target for many pathologies is currently a very interesting area of study with great potential for the scientific community.
Introduction: In this section the authors briefly describe the NAFLD pathology.
The objective of the study is not found.
Methodology. This section does not exist. It is assumed that the authors have carried out a systematic review, as this is how the results are presented. However, essential aspects of the methodology that should be included are missing: the methodology itself (process of collecting and selecting articles), inclusion and exclusion criteria, databases consulted, Boolean markers used, etc.
On the other hand, there is a lack of novel studies on this particular topic. Doing a quick search, I have been able to identify several articles that could have been included in this study to give it value. However, not knowing the criteria used, it is not possible to confirm this.
In addition, it is apparent that very few studies in human models are included. Much progress has been made in this area in recent years, and it is possible to find novel studies with promising results and conclusions. It would be good to be able to integrate them.
Results: The results table lacks important information such as the type of study in each case, as well as the methodology used in each of them (control-placebo...), the number of participants in each case (only the total number of participants is presented).
Discussion. This section does not exist. It is important to discuss the results obtained with previous studies in this line of research. Even if it is a systematic review, the results should always be discussed.
The conclusions should respond to the objective of the study. However, it is not possible to know if this has been done as the objective(s) have not been included.
Round 2
Reviewer 2 Report
I sincerely thank the authors for their careful revision of the manuscript, highlihting the methodolofy of the review, illustrating the text with figures and thorough explanations for the raised issues. The clarifications and additions significantly enhanced the consistency and coherency of the manuscript.
There are minor issues:
Figure legends. Please provide the reference, if the figure was adapted to some other source.
Nomenclature of the lactobacilli. The official nomenclature of lactobaclilli have changed in 2020. The current official names should also be stated (could be in brackets). More information of standing in nomenclature can be found at LPSN page. For example, Lactobacillus reuteri is now classified as Limnosilactobacillus reuteri. https://lpsn.dsmz.de/species/limosilactobacillus-reuteri
Reviewer 3 Report
Dear authors,
Thank you for all changes included. Now the document is more clearly and better estructured.
Author Response
Dear Reviewer, We appreciate the time and effort that you dedicated to providing your valuable feedback and insightful comments on the manuscript. Thank you. Kindest regards, The authors.